# Divergence-Augmented Policy Optimization

**Qing Wang** [*]
Huya AI
Guangzhou, China

**Yingru Li**
The Chinese University of Hong Kong
Shenzhen, China

**Jiechao Xiong**
Tencent AI Lab
Shenzhen, China

**Tong Zhang**
The Hong Kong University of Science and Technology
Hong Kong, China

## Abstract

In deep reinforcement learning, policy optimization methods need to deal with issues such as function approximation and the reuse of off-policy data. Standard policy gradient methods do not handle off-policy data well, leading to premature convergence and instability. This paper introduces a method to stabilize policy optimization when off-policy data are reused. The idea is to include a Bregman divergence between the behavior policy that generates the data and the current policy to ensure small and safe policy updates with off-policy data. The Bregman divergence is calculated between the state distributions of two policies, instead of only on the action probabilities, leading to a divergence augmentation formulation. Empirical experiments on Atari games show that in the data-scarce scenario where the reuse of off-policy data becomes necessary, our method can achieve better performance than other state-of-the-art deep reinforcement learning algorithms.

## 1 Introduction

In recent years, many algorithms based on policy optimization have been proposed for deep reinforcement learning (DRL), leading to great successes in Go, video games, and robotics (Silver et al., 2016; Mnih et al., 2016; Schulman et al., 2015, 2017b). Real-world applications of policy-based methods commonly involve function approximation and data reuse. Typically, the reused data are generated with an earlier version of the policy, leading to off-policy learning. It is known that these issues may cause premature convergence and instability for policy gradient methods (Sutton et al., 2000; Sutton and Barto, 2017).

A standard technique that allows policy optimization methods to handle off-policy data is to use importance sampling to correct trajectories from the behavior policy that generates the data to the target policy (e.g. Retrace (Munos et al., 2016) and V-trace (Espeholt et al., 2018)). The efficiency of these methods depends on the divergence between the behavior policy and the target policy. Moreover, to improve stability of training, one may introduce a regularization term (e.g. Shannon-Gibbs entropy in (Mnih et al., 2016)), or use a proximal objective of the original policy gradient loss (e.g. clipping in (Schulman et al., 2017b; Wang et al., 2016a)). Although the well-adopted method of entropy regularization can stabilize the optimization process (Mnih et al., 2016), this additional entropy regularization alters the learning objective, and prevent the algorithm from converging to the optimal action for each state. Even for the simple case of bandit problems, the monotonic diminishing regularization may fail to converge to the best arm (Cesa-Bianchi et al., 2017).

In this work, we propose a method for policy optimization by adding a Bregman divergence term, which leads to more stable and sample efficient off-policy learning. The Bregman divergence

---

[*]The work was done when the first author was at Tencent AI Lab.

constraint is widely used to explore and exploit optimally in mirror descent methods (Nemirovsky and Yudin, 1983), in which specific form of divergence can attain the optimal rate of regret (sample efficiency) for bandit problems (Audibert et al., 2011; Bubeck and Cesa-Bianchi, 2012). In contrast to the traditional approach of constraining the divergence between target policy and behavior policy conditioned on each state (Schulman et al., 2015), we consider the divergence over the joint state-action space. We show that the policy optimization problem with Bregman divergence on state-action space is equivalent to the standard policy gradient method with divergence-augmented advantage. Under this view, the divergence-augmented policy optimization method not only considers the divergence on the current state but also takes into account the discrepancy of policies on future states, thus can provide a better constraint on the change of policy and encourage "deeper" exploration.

We experiment with the proposed method on the commonly used Atari 2600 environment from Arcade Learning Environment (ALE) (Bellemare et al., 2013). Empirical results show that divergence-augmented policy optimization method performs better than the state-of-the-art algorithm under data-scarce scenarios, i.e., when the sample generating speed is limited and samples in replay memory are reused multiple times. We also conduct a comparative study for the major effect of improvement on these games.

The article is organized as follows: we give the basic background and notations in Section 2. The main method of divergence-augmented policy optimization is presented in Section 3, with connections to previous works discussed in Section 4. Empirical results and studies can be found in Section 5. We conclude this work with a short discussion in Section 6.

## 2 Preliminaries

In this section, we state the basic definition of the Markov decision process considered in this work, as well as the Bregman divergence used in the following discussions.

### 2.1 Markov Decision Process

We consider a Markov decision process (MDP) with infinite-horizon and discounted reward, denoted by $\mathcal{M} = (\mathcal{S}, \mathcal{A}, P, r, d_0, \gamma)$, where $\mathcal{S}$ is the finite state space, $\mathcal{A}$ is the finite action space, $P : \mathcal{S} \times \mathcal{A} \to \Delta(\mathcal{S})$ is the transition function, where $\Delta(\mathcal{S})$ means the space of all probability distributions on $\mathcal{S}$. A reward function is denoted by $r : \mathcal{S} \times \mathcal{A} \to \mathbb{R}$. The distribution of initial state $s_0$ is denoted by $d_0 \in \Delta(\mathcal{S})$. And a discount factor is denoted by $\gamma \in (0, 1)$.

A stochastic policy is denoted by $\pi : \mathcal{S} \to \Delta(\mathcal{A})$. The space of all policies is denoted by $\Pi$. We use the following standard notation of state-value $V^\pi(s_t)$, action-value $Q^\pi(s_t, a_t)$ and advantage $A^\pi(s_t, a_t)$, defined as $V^\pi(s_t) = \mathbb{E}_{\pi|s_t} \sum_{l=0}^\infty \gamma^l r(s_{t+l}, a_{t+l})$, $Q^\pi(s_t, a_t) = \mathbb{E}_{\pi|s_t, a_t} \sum_{l=0}^\infty \gamma^l r(s_{t+l}, a_{t+l})$, and $A^\pi(s_t, a_t) = Q^\pi(s_t, a_t) - V^\pi(s_t)$, where $\mathbb{E}_{\pi|s_t}$ means $a_l \sim \pi(a|s_l), s_{l+1} \sim P(s_{l+1}|s_l, a_l), \forall l \geq t$, and $\mathbb{E}_{\pi|s_t, a_t}$ means $s_{l+1} \sim P(s_{l+1}|s_l, a_l), a_{l+1} \sim \pi(a|s_{l+1}), \forall l \geq t$. We also define the space of policy-induced state-action distributions under $\mathcal{M}$ as

$$\Delta_\Pi = \{\mu \in \Delta(\mathcal{S} \times \mathcal{A}) : \sum_{a'} \mu(s', a') = (1 - \gamma)d_0(s') + \gamma \sum_{s,a} P(s'|s, a)\mu(s, a), \forall s' \in \mathcal{S}\} \quad (1)$$

We use the notation $\mu_\pi$ for the state-action distribution induced by $\pi$. On the other hand, for each $\mu \in \Delta_\Pi$, there also exists a unique policy $\pi_\mu(a|s) = \frac{\mu(s,a)}{\sum_b \mu(s,b)}$ which induces $\mu$. We define the state distribution $d_\pi$ as $d_\pi(s) = (1 - \gamma)\mathbb{E}_{\tau|\pi} \sum_{t=0}^\infty \gamma^t \mathbf{1}(s_t = s)$. Then we have $\mu_\pi(s, a) = d_\pi(s)\pi(a|s)$. We sometimes write $\pi_{\mu_t}$ as $\pi_t$ and $d_{\pi_t}$ as $d_t$ when there is no ambiguity.

In this paper, we mainly focus on the performance of a policy $\pi$ defined as

$$J(\pi) = (1 - \gamma)\mathbb{E}_{\tau|\pi} \sum_{t=0}^\infty \gamma^t r(s_t, a_t) = \mathbb{E}_{d_\pi, \pi} r(s, a) \quad (2)$$

where $\mathbb{E}_{\tau|\pi}$ means $s_0 \sim d_0, a_t \sim \pi(a_t|s_t), s_{t+1} \sim P(s_{t+1}|s_t, a_t), t \geq 0$. We use the notation $\mathbb{E}_{d,\pi} = \mathbb{E}_{s \sim d(\cdot), a \sim \pi(\cdot|s)}$ for brevity.

## 2.2 Bregman Divergence

We define Bregman divergence (Bregman, 1967) as follows (e.g. Definition 5.3 in (Bubeck and Cesa-Bianchi, 2012)). For $\mathcal{D} \subset \mathbb{R}^d$ an open convex set, the closure of $\mathcal{D}$ as $\bar{\mathcal{D}}$, we consider a *Legendre* function $F : \bar{\mathcal{D}} \to \mathbb{R}$ defined as (1) $F$ is strictly convex and admits continuous first partial derivatives on $\mathcal{D}$, and (2) $\lim_{x \to \bar{\mathcal{D}} \backslash \mathcal{D}} \|\nabla F\| = +\infty$. For function $F$, we define the Bregman divergence $D_F : \bar{\mathcal{D}} \times \mathcal{D} \to \mathbb{R}$ as

$$D_F(x, y) = F(x) - F(y) - \langle \nabla F(y), x - y \rangle.$$

The inner product is defined as $\langle x, y \rangle = \sum_i x_i y_i$. For $\mathcal{K} \subset \bar{\mathcal{D}}$ and $\mathcal{K} \cap \mathcal{D} \neq \emptyset$, the Bregman projection

$$z = \arg\min_{x \in \mathcal{K}} D_F(x, y)$$

exists uniquely for all $y \in \mathcal{D}$. Specifically, for $F(x) = \sum_i x_i \log(x_i) - \sum_i x_i$, we recover the Kullback-Leibler (KL) divergence as

$$D_{\text{KL}}(\mu', \mu) = \sum_{s,a} \mu'(s, a) \log \frac{\mu'(s, a)}{\mu(s, a)}$$

for $\mu, \mu' \in \Delta(\mathcal{S} \times \mathcal{A})$ and $\pi, \pi' \in \Pi$. To measure the distance between two policies $\pi$ and $\pi'$, we also use the symbol for conditional "Bregman divergence"[2] associated with state distribution $d$ denoted as

$$D_F^d(\pi', \pi) = \sum_s d(s) D_F(\pi'(\cdot|s), \pi(\cdot|s)). \tag{3}$$

# 3 Method

In this section, we present the proposed method from the motivation of mirror descent and then discuss the parametrization and off-policy correction we employed in the practical learning algorithm.

## 3.1 Policy Optimization and Mirror Descent

The mirror descent (MD) method (Nemirovsky and Yudin, 1983) is a central topic in the optimization and online learning research literature. As a first-order method for optimization, the mirror descent method can recover several interesting algorithms discovered previously (Sutton et al., 2000; Kakade, 2002; Peters et al., 2010; Schulman et al., 2015). On the other hand, as an online learning method, the online (stochastic) mirror descent method can achieve (near-)optimal sample efficiency for a wide range of problems (Audibert and Bubeck, 2009; Audibert et al., 2011; Zimin and Neu, 2013). In this work, following a series of previous works (Zimin and Neu, 2013; Neu et al., 2017), we investigate the (online) mirror descent method for policy optimization. We denote the state-action distribution at iteration $t$ as $\mu_t$, and $\ell_t(\mu) = \langle g_t, \mu \rangle$ as the linear loss function for $\mu$ at iteration $t$. Without otherwise noted, we consider the negative reward as the loss objective $\ell_t(\mu) = -\langle r, \mu \rangle$, which also corresponds to the policy performance $\ell_t(\mu) \equiv -J(\pi_\mu)$ by Formula (2). We consider the mirror map method associated with Legendre function $F$ as

$$\nabla F(\tilde{\mu}_{t+1}) = \nabla F(\mu_t) - \eta g_t \tag{4}$$
$$\mu_{t+1} \in \Pi_{\Delta_\Pi}(\tilde{\mu}_{t+1}), \tag{5}$$

where $\tilde{\mu}_{t+1} \in \Delta(\mathcal{S} \times \mathcal{A})$ and $g_t = \nabla \ell_t(\mu_t)$. It is well-known (Beck and Teboulle, 2003) that an equivalent formulation of mirror map (4) is

$$\mu_{t+1} = \arg\min_{\mu \in \Delta_\Pi} D_F(\mu, \tilde{\mu}_{t+1}) \tag{6}$$
$$= \arg\min_{\mu \in \Delta_\Pi} D_F(\mu, \mu_t) + \eta \langle g_t, \mu \rangle, \tag{7}$$

The former formulation (6) takes the view of non-linear sub-gradient projection in convex optimization, while the later formulation (7) can be interpreted as a regularized optimization and is the usual definition of mirror descent (Nemirovsky and Yudin, 1983; Beck and Teboulle, 2003; Bubeck, 2015). In this work, we will mostly investigate the approximate algorithm in the later formulation (7).

## 3.2 Parametric Policy-based Algorithm

In the mirror descent view for policy optimization on state-action space as in Formula (7), we need to compute the projection of $\mu$ onto the space of $\Delta_\Pi$. For the special case of KL-divergence on $\mu$, the sub-problem of finding minimum in (7) can be done efficiently, assuming the knowledge of transition function $P$ (See Proposition 1 in (Zimin and Neu, 2013)). However, for a general divergence and real-world problems with unknown transition matrices, the projection in (7) is non-trivial to implement. In this section, we consider direct optimization in the (parametric) policy space without explicit projection. Specifically, we consider $\mu_\pi$ as a function of $\pi$, and $\pi$ parametrized as $\pi_\theta$. The Formula (7) can be written as

$$\pi_{t+1} = \arg\min_{\pi} D_F(\mu_\pi, \mu_t) + \eta \langle g_t, \mu_\pi \rangle. \tag{8}$$

Instead of solving globally, we approximate Formula (8) with gradient descent on $\pi$. From the celebrated policy gradient theorem (Sutton et al., 2000), we have the following lemma:

**Lemma 1.** *(Policy Gradient Theorem (Sutton et al., 2000)) For $d_\pi$ and $\mu_\pi$ defined previously, the following equation holds for any state-action function $f : \mathcal{S} \times \mathcal{A} \to \mathbb{R}$:*

$$\sum_{s,a} f(s,a) \nabla_\theta \mu_\pi(s,a) = \sum_{s,a} d_\pi(s) \mathcal{Q}^\pi(f)(s,a) \nabla_\theta \pi(a|s),$$

*where $\mathcal{Q}^\pi$ is defined as an operator such that*

$$\mathcal{Q}^\pi(f)(s,a) = \mathbb{E}_{\pi|s_t=s,a_t=a} \sum_{l=0}^{\infty} \gamma^l f(s_{t+l}, a_{t+l}).$$

Decomposing the loss and divergence in two parts (8), we have

$$\nabla_\theta \langle g_t, \mu_\pi \rangle = \langle d_\pi \mathcal{Q}^\pi(g_t), \nabla_\theta \pi(a|s) \rangle, \tag{9}$$

which is the usual policy gradient, and

$$\nabla_\theta D_F(\mu_\pi, \mu_t) = \langle \nabla F(\mu_\pi) - \nabla F(\mu_t), \nabla_\theta \mu_\pi \rangle = \langle d_\pi \mathcal{Q}^\pi (\nabla F(\mu_\pi) - \nabla F(\mu_t)), \nabla_\theta \pi(a|s) \rangle. \tag{10}$$

Similarly, we have the policy gradient for the conditional divergence (3) as

$$\nabla_\theta D_F^{d_t}(\pi, \pi_t) = \langle d_t(\nabla F(\pi) - \nabla F(\pi_t)), \nabla_\theta \pi(a|s) \rangle,$$

which does not have a discounted sum, since $d_t$ is fixed and independent of $\pi = \pi_\mu$.

## 3.3 Off-policy Correction

In this section, we discuss the practical method for estimating $\mathcal{Q}^\pi(f)$ under a behavior policy $\pi_t$. In distributed reinforcement learning with asynchronous gradient update, the policy $\pi_t$ which generated the trajectories may deviate from the policy $\pi_\theta$ currently being optimized. Thus off-policy correction is usually needed for the robustness of the algorithm (e.g. V-trace as in IMPALA (Espeholt et al., 2018)). Consider

$$\sum_{s,a} d_\pi(s) \mathcal{Q}^\pi(f)(s,a) \nabla_\theta \pi(a|s) = \mathbb{E}_{(s,a) \sim \pi d_\pi} \mathcal{Q}^\pi(f)(s,a) \nabla_\theta \log \pi(a|s)$$

$$= \mathbb{E}_{(s,a) \sim \pi_t d_{\pi_t}} \frac{d_\pi(s)}{d_{\pi_t}(s)} \frac{\pi(a|s)}{\pi_t(a|s)} \mathcal{Q}^\pi(f)(s,a) \nabla_\theta \log \pi(a|s)$$

for $f = g_t$ or $f = \nabla F(\mu_\pi) - \nabla F(\mu_t)$. We would like to have an accurate estimation of $\mathcal{Q}^\pi(g_t)$ (9) and $\mathcal{Q}^\pi(\nabla F(\mu_\pi) - \nabla F(\mu_t))$ (10), and correct the deviation from $d_{\pi_t}$ to $d_\pi$ and $\pi_t$ to $\pi$.

For the estimation of $\mathcal{Q}^\pi(f)$ under a behavior policy $\pi_t$, possible methods include Retrace (Munos et al., 2016) providing an estimator of state-action value $\mathcal{Q}^\pi(f)$, and V-trace (Espeholt et al., 2018) providing an estimator of state value $\mathbb{E}_{a \sim \pi} \mathcal{Q}^\pi(f)(s,a)$. In this work, we utilize the V-trace (Section 4.1 (Espeholt et al., 2018)) estimation $v_{s_i} = v_i$ along a trajectory starting at $(s_i, a_i = s, a)$ under $\pi_t$.

Details of multi-step Q-value estimation can be found in Appendix A. With the value estimation $v_s$, the $\mathcal{Q}^\pi(g_t)$ is estimated with

$$\hat{A}_{s,a} = r_i + \gamma v_{i+1} - V_\theta(s_i). \tag{11}$$

We subtract a baseline $V_\theta(s_i)$ to reduce variance in estimation, as $\mathbb{E}_{\pi_t, d_t} \frac{\pi_\theta}{\pi_t} V_\theta(s) \nabla_\theta \log \pi_\theta = 0$. For the estimation of $\mathcal{Q}^\pi(\nabla F(\mu_\pi) - \nabla F(\mu_t))$, we use the $n$-steps truncated importance sampling as

$$\hat{D}_{s,a} = f(s_i, a_i) + \sum_{j=1}^{n} \gamma^j (\prod_{k=0}^{j-1} c_{i+k}) \rho_{i+j} f(s_{i+j}, a_{i+j}). \tag{12}$$

in which we use the notation $c_j = \min(\bar{c}_D, \frac{\pi_\theta(a_j|s_j)}{\pi_t(a_j|s_j)})$ and $\rho_j = \min(\bar{\rho}_D, \frac{\pi_\theta(a_j|s_j)}{\pi_t(a_j|s_j)})$. The formula also corresponds to V-trace under the condition $V(\cdot) \equiv 0$. For RNN model trained on continuous roll-out samples, we set $n$ equals to the max-length till the end of roll-out.

For the correction of state distribution $d_\pi(s)/d_{\pi_t}(s)$, previous solutions include the use of emphatic algorithms as in (Sutton et al., 2016), or through an estimate of state density ratio as in (Liu et al., 2018). However, in our experience, the estimation of density ratio will introduce additional error, which may lead to worse performance in practice. Therefore in this paper, we propose a different solution by restricting our attention to the correction of $\pi_t$ to $\pi$ via importance sampling and omitting the difference of $d_\pi/d_{\pi_t}$ in the algorithm. This introduces a bias in the gradient estimation, which we propose a new method to handle in this paper. Specifically, we show that although the omission of the state ratio introduces a bias in the gradient, the bias can be bounded by the regularization term of conditional KL divergence (see Appendix B). Therefore by explicitly adding an KL divergence regularization, we can effectively control the degree of off-policy bias caused by $d_\pi/d_{\pi_t}$ in that small regularization value implies a small bias. This approach naturally combines mirror descent with KL divergence regularization, leading to a more stable algorithm that is robust to off-policy data, as we will demonstrate by empirical experiments.

The final loss consists of the policy loss $L_\pi(\theta)$ and the value loss $L_v(\theta)$. To be specific, the gradient of policy loss is defined as

$$\nabla_\theta L_\pi(\theta) = \mathbb{E}_{\pi_t, d_t} \frac{\pi}{\pi_t} (\hat{D}_{s,a} - \eta \hat{A}_{s,a}) \nabla_\theta \log \pi. \tag{13}$$

We can also use proximal methods like PPO (Schulman et al., 2017b) in conjunction with divergence augmentation. A practical implementation is elaborated later in Formula (19). In addition to the policy loss, we also update $V_\theta$ with value gradient defined as

$$\nabla L_v(\theta) = \mathbb{E}_{\pi_t, d_t} \frac{\pi}{\pi_t} (V_\theta(s) - v_s) \nabla_\theta V_\theta(s), \tag{14}$$

where $v_s = v_{s_i}$ is the multi-step value estimation with V-trace. The parameter $\theta$ is then updated with a mixture of policy loss and value loss

$$\theta \leftarrow \theta - \alpha_t (\nabla_\theta L_\pi(\theta) + b \nabla_\theta L_v(\theta)), \tag{15}$$

in which $\alpha_t$ is the current learning rate, and $b$ is the loss scaling coefficient. The algorithm is summarized in Algorithm 1.

## 4 Related Works

The policy performance in Equation (2) and the well-known policy difference lemma (Kakade and Langford, 2002) serve a fundamental role in policy-based reinforcement learning (e.g TRPO, PPO (Schulman et al., 2015, 2017b)). The gradient with respect to the policy performance and policy difference provides a natural direction for policy optimization. And to restrict the changes in each policy improvement step, as well as encouraging exploration at the early stage, the constraint-based policy optimization methods try to limit the changes in the policy by constraining the divergence between behavior policy and current policy. The use of entropy maximization in reinforcement learning can be dated back to the work of Williams and Peng (1991). And methods with relative entropy regularization include Peters et al. (2010); Schulman et al. (2015). The relationship between these methods and the mirror descent method has been discussed in Neu et al. (2017). With

---

**Algorithm 1** Divergence-Augmented Policy Optimization (DAPO)

---

**Input:** $D_F(\mu', \mu)$, total iteration $T$, batch size $M$, learning rate $\alpha_t$.
**Initialize :** randomly initiate $\theta_0$
**for** $t = 0$ **to** $T$ **do**
  (in parallel) Use $\pi_t = \pi_{\theta_t}$ to generate trajectories.
  **for** $m = 1$ **to** $M$ **do**
    Sample $(s_i, a_i) \in \mathcal{S} \times \mathcal{A}$ w.p. $d_t \pi_t$.
    Estimate state value $v_{s_i}$ (e.g. by V-trace).
    Calculate Q-value estimation $\hat{A}_{s,a}$ (11) and divergence estimation $\hat{D}_{s,a}$ (12).
      $\hat{A}_{s,a} = r_i + \gamma v_{i+1} - V_\theta(s_i),$
      $\hat{D}_{s,a} = f(s_i, a_i) + \sum_{j=1}^{n} \gamma^j (\prod_{k=0}^{j-1} c_{i+k}) \rho_{i+j} f(s_{i+j}, a_{i+j}).$
    Update $\theta$ with respect of policy loss (13, optionally 19) and value loss (14)
      $\theta \leftarrow \theta - \alpha_t(\nabla_\theta L_\pi(\theta) + b \nabla_\theta L_v(\theta)).$
  **end for**
  Set $\theta_{t+1} = \theta$.
**end for**

---

notations in this work, consider the natural choice of $F$ as the *negative Shannon entropy* defined as $F(x) = \sum_i x_i \log(x_i)$, the $D_F(\cdot, \cdot)$ becomes the KL-divergence $D_{\text{KL}}(\cdot, \cdot)$. By the equivalence of sub-gradient projection (6) and mirror descent (7), the mirror descent policy optimization with KL-divergence can be written as

$$\mu_{t+1} = \arg\min_{\mu \in \Delta_\Pi} D_{\text{KL}}(\mu, \tilde{\mu}_{t+1}) = \arg\min_{\mu \in \Delta_\Pi} D_{\text{KL}}(\mu, \mu_t) + \eta \langle g_t, \mu \rangle. \tag{16}$$

Under slightly different settings, this learning objective is the regularized version of the constrained optimization problem considered in Relative Entropy Policy Search (REPS) (Peters et al., 2010); And for $\ell_t(\mu)$ depending on $t$, the Equation (16) can also recover the O-REPS method considered in Zimin and Neu (2013). On the other hand, as the KL-divergence (and Bregman divergence) is asymmetric, we can also replace the $D_F(x, y)$ in either formulation (6, 7) with reverse KL $D_{\text{KL}}(y, x)$, which will result in different iterative algorithms (as the reverse KL is no longer a Bregman divergence, the equivalence of Formula (6) and (7) no longer holds). Consider replacing $D_F(\mu, \tilde{\mu}_{t+1})$ with $D_{\text{KL}}(\tilde{\mu}_{t+1}, \mu)$ in sub-gradient projection (6), we have the "mirror map" method with reverse KL as

$$\mu_{t+1} = \arg\min_{\mu \in \Delta_\Pi} D_{\text{KL}}(\tilde{\mu}_{t+1}, \mu), \tag{17}$$

which is essentially the MPO algorithm (Abdolmaleki et al., 2018) under a probabilistic inference perspective, and MARWIL algorithm (Wang et al., 2018) when learning from off-policy data. Similarly, consider the replacement of $D_F(\mu, \mu_t)$ with $D_{\text{KL}}(\mu_t, \mu)$ in mirror descent (7), we have the "mirror descent" method with reverse KL as

$$\mu_{t+1} = \arg\min_{\mu \in \Delta_\Pi} D_{\text{KL}}(\mu_t, \mu) + \eta \langle g_t, \mu \rangle, \tag{18}$$

which can approximately recover the TRPO optimization objective (Schulman et al., 2015) (if the relative entropy between two state-action distributions $D_{\text{KL}}(\mu_t, \mu)$ in (18) is replaced by the conditional entropy $D_{\text{KL}}^{d_t}(\pi_t, \pi)$, also see Section 5.1 of Neu et al. (2017)).

Besides, we note that there are other choices of constraint for policy optimization as well. For example, in (Lee et al., 2018; Chow et al., 2018; Lee et al., 2019), a Tsallis entropy is used to promote sparsity in the policy distribution. And in (Belousov and Peters, 2017), the authors generalize KL, Hellinger distance, and reversed KL to the class of $f$-divergence. In preliminary results, we found divergence based on 0-potential (Audibert et al., 2011; Bubeck and Cesa-Bianchi, 2012) is also promising for policy optimization. We left this for future research.

For multi-step KL divergence regularized policy optimization, we note that the formulation also corresponds to the KL-divergence-augmented return considered previously in several works (Fox et al. (2015), Section 3 of Schulman et al. (2017a)), although in Schulman et al. (2017a) the authors use a fixed behavior policy instead of $\pi_t$ as in ours. More often, the Shannon-entropy-augmented return can be dated back to earlier works (Kappen, 2005; Todorov, 2007; Ziebart et al., 2008; Nachum et al., 2017), and is a central topic in "soft" reinforcement learning (Haarnoja et al., 2017, 2018).

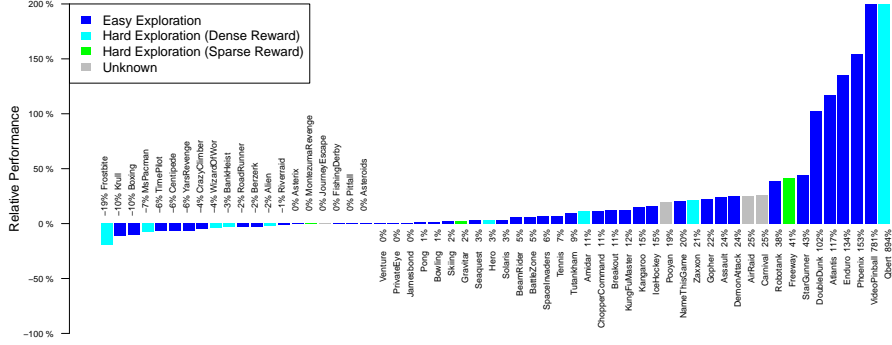

Figure 1: Relative score improvement of PPO+DA compared with PPO on 58 Atari environments. The relative performance is calculated as a $\frac{\text{proposed}-\text{baseline}}{\max(\text{human},\text{baseline})-\text{random}}$ (Wang et al., 2016b). The Atari games are categorized according to Figure 4 of (Oh et al., 2018).

The mirror descent method is originally introduced by the seminal work of Nemirovsky and Yudin (1983) as a convex optimization method. Also, the online stochastic mirror descent method has alternative views, e.g. Follow the Regularized Leader (McMahan, 2011), and Proximal Point Algorithm (Rockafellar, 1976). For more discussions on mirror descent and online learning, we refer interested readers to the work of Cesa-Bianchi and Lugosi (2006) and Bubeck and Cesa-Bianchi (2012).

## 5 Experiments

In the experiments, we test the exploratory effect of divergence augmentation comparing with entropy augmentation, and the empirical difference between multi-step and 1-step divergence. For the experiments, we mainly consider the DAPO algorithm (1) associated with the conditional KL divergence (see $R_C$ and $D_C$ in (Neu et al., 2017)). For $F(\mu) = \sum_{s,a} \mu(s,a) \log \frac{\mu(s,a)}{\sum_b \mu(s,b)}$, we have the gradient in (10) as

$$\nabla F(\mu_\pi) - \nabla F(\mu_t) = \log \frac{\pi}{\pi_t}.$$

The multi-step divergence augmentation term as in (12) is then calculated as

$$\hat{D}_{s,a}^{\text{KL}} = \log \frac{\pi(a_i|s_i)}{\pi_t(a_i|s_i)} + \sum_{j=1}^{n} \gamma^j \left( \prod_{k=1}^{j-1} c_{i+k} \right) \rho_{i+j} \log \frac{\pi(a_{i+j}|s_{i+j})}{\pi_t(a_{i+j}|s_{i+j})}.$$

As a baseline, we also implement the PPO algorithm with a V-trace (Espeholt et al., 2018) estimation of advantage function $A^\pi$ for target policy[3]. Specifically, we consider the policy loss as:

$$L_\pi^{\text{PPO}}(\theta) = \mathbb{E}_{\pi_t, d_t} \min\left( \frac{\pi_\theta}{\pi_t} A_{s,a}, \text{clip}\left( \frac{\pi_\theta}{\pi_t}, 1 - \epsilon, 1 + \epsilon \right) A_{s,a} \right), \tag{19}$$

where we choose $\epsilon = 0.2$ and the advantage is estimated by $R_{s,a}$. We also tested the DAPO algorithm with PPO, with the advantage estimation $A_{s,a}$ in (19) replaced with $\hat{A}_{s,a} - \frac{1}{\eta} \hat{D}_{s,a}$ defined in (11) and (12). We will refer to this algorithm as PPO+DA in the following sections.

### 5.1 Algorithm Settings

The algorithm is implemented with TensorFlow (Abadi et al., 2016). For efficient training with deep neural networks, we use the Adam (Kingma and Ba, 2014) method for optimization. The learning rate is linearly scaled from 1e-3 to 0. The parameters are updated according to a mixture of policy loss and value loss, with the loss scaling coefficient $c = 0.5$. In calculating multi-step $\lambda$-returns $R_{s,a}$

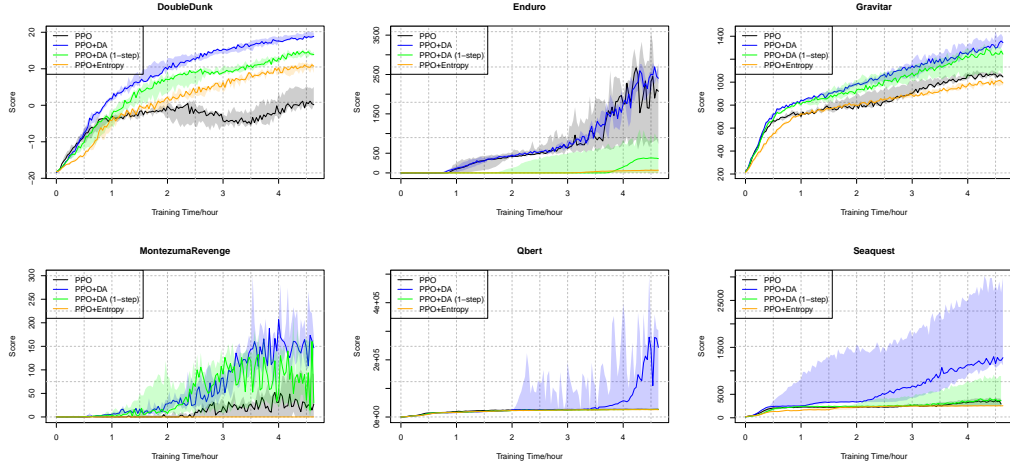

Figure 2: Performance comparison of selected environments of Atari games. The performance of **PPO**, PPO+DA, PPO+DA (1-step), and PPO+Entropy are plotted in different colors. The score for each game is plotted on the y-axis with running time on the x-axis, as the algorithm is paralleled asynchronously in a distributed environment. For each line in the plots, we run the experiment 5 times with the same parameters and environment settings. The median scores are plotted in solid lines, while the regions between 25% and 75% quantiles are shaded with respective colors.

and divergence $D_{s,a}$, we use fixed $\lambda = 0.9$ and $\gamma = 0.99$. The batch size is set to 1024, with roll-out length set to 32, resulting in 1024/32=32 roll-outs in a batch. The policy $\pi_t$ and value $V_t$ is updated every 100 iterations ($M = 100$ in Algorithm 1). With our implementation, the training speed is about 25k samples per second, and the data generating speed is about 220 samples per second for each actor, resulting in about 3500 samples per second for a total of 16 actors. Note that the PPO results may not be directly comparable with other works (Schulman et al., 2017b; Espeholt et al., 2018; Xu et al., 2018), mainly due to the different number of actors used. Unless otherwise noted, each experiment is allowed to run 16000 seconds (about 4.5 hours), corresponding a total of 60M samples generated and 400M samples (with replacement) trained. Details of experimental settings can be found in Appendix A.

## 5.2 Empirical Results

We test the algorithm on 58 Atari environments and calculate its relative performance with PPO (Schulman et al., 2017b). The empirical performance is plotted in Figure 1. We run PPO and PPO+DA with the same environmental settings and computational resources. The relative performance is calculated as $\frac{\text{proposed} - \text{baseline}}{\max(\text{human,baseline}) - \text{random}}$ (Wang et al., 2016b). We also categorize the game environments into easy exploration games and hard exploration games (Oh et al., 2018). We see that with a KL-divergence-augmented return, the algorithm PPO+DA performs better than the baseline method, especially for the games that may have local minimums and require deeper exploration. We plot the learning curves of PPO+DA (in blue) comparing with PPO (in **black**) and other baseline methods on 6 typical environments in Figure 2. Detailed learning curves for PPO and PPO+DA for the complete 58 games can be found in Figure 3 in the Appendix.

### 5.2.1 Divergence augmentation vs Entropy augmentation

We test the effect of divergence augmentation in contrast to the entropy augmentation (plotted in orange in Figure 2). Entropy augmentation can prevent premature convergence and encourage exploration as well as stabilize policy during optimization. However, the additional entropy may hinder the convergence to the optimal action, as it alters the original learning objective. We set $f(s,a)$ as $\log \pi(a|s)$ in Formula (12), and experiment the algorithm with $\frac{1}{\eta} = 0.5, 0.1, 0.01, 0.001$, in which we found that $\frac{1}{\eta} = 0.1$ performs best. From the empirical results, we see that divergence-augmented PPO works better, while the entropy-augmented version may be too conservative on policy changes, resulting in inferior performance on these games.

### 5.2.2 Multi-step divergence vs 1-step divergence

In Figure 2, we also test the PPO+DA algorithm with its 1-step divergence-augmented counterpart (plotted in green). We rerun the experiments with the parameter $\bar{c}_D$ (Formula (12)) set to 0, which means we only aggregate the divergence on the current state and action $f(s_i, a_i)$, without summing up future discounted divergence $f(s_{i+j}, a_{i+j})$. This method also relates to the conditional divergence defined in Formula (3), and shares more similarities with previous works on regularized and constrained policy optimization methods (Schulman et al., 2015; Achiam et al., 2017). We see that with multi-step divergence augmentation, the algorithm can achieve high scores, especially on games requiring deeper exploration like Enduro and Qbert. We hypothesize that the accumulated divergence on future states can encourage the policy to explore more efficiently.

## 6 Conclusion

In this paper, we proposed a divergence-augmented policy optimization method to improve the stability of policy gradient methods when it is necessary to reuse off-policy data. We showed that the proposed divergence augmentation technique can be viewed as imposing Bregman divergence constraint on the state-action space, which is related to online mirror descent methods. Experiments on Atari games showed that in the data-scarce scenario, the proposed method works better than other state-of-the-art algorithms such as PPO. Our results showed that the technique of divergence augmentation is effective when data generated by previous policies are reused in policy optimization.

## Footnotes

[2]Note that $D_F^d$ may not be a Bregman divergence.

[3]In the original PPO (Schulman et al., 2017b) they use $\hat{A}$ as the advantage estimation of behavior policy $A^{\pi_t}$.

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
