[Supplementary Material · Divergence_Augmented_Policy_Optimization.pdf]

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

# A    Details of the algorithm

## A.1    Environment Settings

We evaluate the algorithm on the Atari 2600 video games from Arcade Learning Environment (ALE) (Bellemare et al., 2013), which is widely used as a standard benchmark for deep reinforcement learning, especially for distributed training (Horgan et al., 2018; Espeholt et al., 2018). There are at least two evaluation protocols, One is the "human starts" protocol (Hasselt et al., 2016), in which each episode for evaluation is started at a state sampled from human play. The other is the "no-ops start" protocol (Mnih et al., 2015), in which the starting state of an episode for evaluation is generated by playing a random length of no-op actions in the environment. We adopt the later evaluation protocol both for training and testing in this work. The environment is based on the OpenAI Gym (Brockman et al., 2016). We mostly follow the environment settings as in (Mnih et al., 2015). The environment is randomly initialized by playing a random number (no more than 30) of no-op actions. During playing, each action is repeated for 4 contiguous frames, and every 4th frame is taken a pixel-wise max over the previous frame, and then returned as the screen observation. The size of the raw screen is 210×160 pixels with 128 colors. The colored image is firstly converted to gray-scale and then resized to 84×84 pixels represented by integers from 0 to 255, followed by a scaling to floats between 0 to 1. We also use the "episodic life" trick in the training phase: For games with a life counter, the loss of life is marked as an end for the current episode. The rewards are clipped with a $sgn()$ function, such that positive rewards are represented by 1, negative rewards as -1, and 0 otherwise. For some games (e.g. Atlantis) we observe that there is a maximum limit of 100000 steps for each episode, corresponding to 400000 raw frames. In addition to the settings above, we also reset the environment if no reward is received in 1000 steps, to prevent the environment from accidental stuck.

## A.2    Network Structure

For comparable results, we use a similar network structure as in (Mnih et al., 2015). The first layer consists of 32 convolution filters of 8×8 with stride 4 and applies a ReLU non-linearity. And the second layer convolves the image with 64 filters of 4×4 with stride 2 followed by a ReLU rectifier. The third layer has 64 filters of 3×3 with stride 1 followed by a rectifier. The final hidden layer is fully-connected and consists of 512 ReLU units. The output layer is two-headed, representing $\pi_\theta(\cdot|s)$ and $V_\theta(s)$ respectively. For the vanilla model, the input is stacked with 4 frames; while for RNN model, we put an additional LSTM (Hochreiter and Schmidhuber, 1997) layer with 256 cells after the fully-connected layer, which is similar to the previous works (Espeholt et al., 2018).

## A.3    Training Framework

For large-scale training with the interested algorithms, we adopt an "Actor-Learner" style (Horgan et al., 2018; Espeholt et al., 2018) distributed framework. In our distributed framework, actors are responsible for generating massive trajectories with current policy; while learners are responsible for updating policy with the data generated by the actors. To be specific, in its main loop, an actor runs a local environment with actions from current local policy and caches the generated data at local memory. The running policy is updated periodically to the latest policy at the learner; while the generated data is sent to the learner asynchronously. At the learner side, the learner keeps at most 20 latest episodes generated by each actor respectively, in a FIFO manner. Each batch of samples for training are randomly sampled from these trajectories with replacement. We deploy the distributed training framework on a small cluster. The learner runs on a GPU machine and occupies an M40 card, while actors run in 16 parallel processes on 2.5GHz Xeon Gold 6133 CPUs.

## A.4    Multi-step Return

In this work, we utilize the V-trace (Section 4.1 (Espeholt et al., 2018)) estimation $v_{s_i} = v_i$ along a trajectory starting at $(s_i, a_i = s, a)$ under $\pi_t$ defined recursively as

$$v_j = \begin{cases} V_\theta(s_j) + \delta_j V_\theta + \gamma c_j(v_{j+1} - V_\theta(s_{j+1})), & i \le j < n \\ r_j + \gamma \lambda_V v_{j+1} + \gamma(1 - \lambda_V)V_t(s_{j+1}), & n \le j < T \end{cases} \tag{20}$$

Table 1: Hyper-parameters

| Name | Value |
|---|---|
| Batch size | 1024 |
| Replay memory size | 16384 ($2^{14}$) |
| $\lambda$ | 0.9 |
| Rollout length | 32 |
| Burn-in samples | 1024 |
| Learning rate | 0.001 to 0 |
| $\bar{c}_D$ (Formula 12) | 0.5 |
| $\bar{\rho}_D$ (Formula 12) | 1.0 |
| $\bar{c}_V$ (Formula 20) | 1.0 |
| $\bar{\rho}_V$ (Formula 20) | 1.0 |
| $\epsilon$ (Formula 19) | 0.2 |
| $1/\eta$ | 0.5 |
| $b$ (Formula 15) | 0.5 |
| Optimizer | Adam |

where $\delta_j V_\theta = \rho_j(r_j + \gamma V_\theta(s_{j+1}) - V_\theta(s_j))$, $\rho_j = \min(\bar{\rho}_V, \frac{\pi_\theta(a_j|s_j)}{\pi_t(a_j|s_j)})$, $c_j = \min(\bar{c}_V, \frac{\pi_\theta(a_j|s_j)}{\pi_t(a_j|s_j)})$, and $0 \leq \lambda_V \leq 1$. The state value estimation function at iteration $t$ is denoted as $V_t(\cdot) = V_{\theta_t}(\cdot)$. The definition of (20) can be seen as following V-trace algorithm along the roll-out (for which we have $\pi_\theta(a_j|s_j)$ and $V_\theta(s_j)$ for $i \leq j < n$, and switch to TD($\lambda$) until a terminal time $T$ (which is estimated offline as we only have $V_t(s_j)$ instead of $\pi_\theta(a_j|s_j)$ and $V_\theta(s_j)$ for $n \leq j < T$). It is noted that TD($\lambda$) also corresponds to V-trace in on-policy settings (Remark 2, (Espeholt et al., 2018)).

### A.5 Hyper-parameters

The default hyper-parameters used in our experiments are given in Table 1.

## B Theoretical Analysis

In this section, we provide some theoretical analysis of the parametrized algorithm considered in this work. We show that the bias introduced by omitting the state ratio can be bounded by the divergence up to a constant factor.

### B.1 Error bound of the biased gradient

Without ambiguity, we define $\pi = \pi_\theta$, $\pi_t = \pi_{\theta_t}$ for brevity, the policy value as

$$V(\theta) = \langle r, \mu_{\pi_\theta} \rangle = \mathbb{E}_{s,a \sim d_\pi \pi} r(s, a),$$

and the conditional KL-divergence for parametrized policies as

$$D(\theta, \theta_t) = \mathbb{E}_{s,a \sim d_\pi \pi} \log \frac{\pi(a|s)}{\pi_t(a|s)}.$$

The algorithm iteratively maximizes the following equation with SGD steps

$$\theta_{t+1} \approx \arg\max_\theta f(\theta, \theta_t) \equiv V(\theta) - \lambda D(\theta, \theta_t). \tag{1}$$

Denote $A^\pi(s, a)$ as the advantage function of reward $r$ following target policy $\pi$, and $\mathbf{A}^\pi_{\pi_t}(s, a)$ as the advantage function of pseudo reward ($\log \pi - \log \pi_t$) following target policy $\pi$, the gradient of (1) is

$$\nabla_\theta f(\theta, \theta_t) = \mathbb{E}_{(s,a) \sim d_{\pi_t} \pi_t} \frac{d_\pi(s)}{d_{\pi_t}(s)} \frac{\pi(a|s)}{\pi_t(a|s)} (A^\pi(s, a) - \lambda \mathbf{A}^\pi_{\pi_t}(s, a)) \nabla_\theta \log \pi(a|s).$$

In the actual implementation, we omit the state ratio of $d_\pi/d_{\pi_t}$, resulting in a biased gradient

$$g(\theta, \theta_t) = \mathbb{E}_{(s,a) \sim d_{\pi_t} \pi_t} \frac{\pi(a|s)}{\pi_t(a|s)} (A^\pi(s, a) - \lambda \mathbf{A}^\pi_{\pi_t}(s, a)) \nabla_\theta \log \pi(a|s).$$

In the following proposition, for target policy $\pi \equiv \pi_\theta$ and reference policy $\tilde{\pi} \equiv \pi_{\tilde{\theta}}$, we show that the error $\delta(\theta, \tilde{\theta}) \equiv \left\| \nabla f(\theta, \tilde{\theta}) - g(\theta, \tilde{\theta}) \right\|$ introduced by omitting the state ratio can be bounded by the conditional KL divergence. To be rigorous, we make the following assumptions:

**Assumption 1** (Universal boundedness). *For all $\theta, \tilde{\theta} \in \Theta$, the set for policy parametrization, there exist non-negative constants $\zeta_1, \zeta_2$ such that,*

$$\max_s \mathbb{E}_{a \sim \pi} \left\| \nabla_\theta \ln \pi(a|s) \right\| \leq \zeta_1,$$

$$\max \left\{ \max_{s,a} |A^\pi(s,a)|, \max_{s,a} |A^\pi(s,a) - \mathbf{A}_{\tilde{\pi}}^\pi(s,a)| \right\} \leq \zeta_2.$$

Then we have the following proposition

**Proposition 1.** *Under Assumption 1, the norm of the gradient bias can be bounded by the conditional KL-divergence:*

$$\delta(\theta, \tilde{\theta})^2 \leq cD(\theta, \tilde{\theta}).$$

*Proof.* The proof provided here is based on the perturbation theory. We firstly define the symbols and notations we used in the proof. Consider the difference on each state denoted as

$$\Delta_{\theta, \tilde{\theta}}(s) = \mathbb{E}_{a \sim \pi} \left\| [A^\pi(s,a) - \lambda \mathbf{A}_{\tilde{\pi}}^\pi(s,a)] \nabla_\theta \ln \pi(a|s) \right\|. \tag{2}$$

By triangular inequality and Hölder's inequality, we have

$$\delta(\theta, \tilde{\theta}) = \left\| \nabla_\theta f(\theta, \tilde{\theta}) - g(\theta, \tilde{\theta}) \right\| \leq \langle |d_\pi - d_{\tilde{\pi}}|, \Delta_{\theta, \tilde{\theta}} \rangle \leq \left\| d_\pi - d_{\tilde{\pi}} \right\|_1 \left\| \Delta_{\theta, \tilde{\theta}} \right\|_\infty.$$

Let $P_\pi \in \mathbb{R}^{|S| \times |S|}$ be the transition matrix associated with policy $\pi$

$$P_\pi(s'|s) = \sum_a P(s'|s,a) \pi(a|s).$$

The discounted state distribution can be written as

$$d_\pi = (1-\gamma) \sum_{t=0}^\infty (\gamma P_\pi)^t d_0 = (1-\gamma)(I - \gamma P_\pi)^{-1} d_0,$$

where $d_0$ is the initial state distribution. For policies $\pi$ and $\tilde{\pi}$, consider the matrices $G \equiv (I - \gamma P_\pi)^{-1}$ and $\tilde{G} \equiv (I - \gamma P_{\tilde{\pi}})^{-1}$, we have

$$G^{-1} - \tilde{G}^{-1} = (I - \gamma P_\pi) - (I - \gamma P_{\tilde{\pi}}) = \gamma(P_{\tilde{\pi}} - P_\pi).$$

Multiplying by $\tilde{G}$ and $G$ on the left and right side respectively, we have

$$\tilde{G} - G = \gamma \tilde{G}(P_{\tilde{\pi}} - P_\pi)G.$$

The difference of state distribution can be bounded as

$$\begin{aligned}
\left\| d_\pi - d_{\tilde{\pi}} \right\|_1 &= \left\| (1-\gamma)(G - \tilde{G})d_0 \right\|_1 \\
&= \left\| \gamma(1-\gamma)\tilde{G}(P_\pi - P_{\tilde{\pi}})Gd_0 \right\|_1 \\
&= \left\| \gamma\tilde{G}(P_\pi - P_{\tilde{\pi}})d_\pi \right\|_1 \\
&\leq \gamma \left\| \tilde{G} \right\|_1 \left\| (P_\pi - P_{\tilde{\pi}})d_\pi \right\|_1 \\
&= \gamma \left\| \sum_{t=0}^\infty (\gamma P_{\tilde{\pi}})^t \right\|_1 \left\| (P_\pi - P_{\tilde{\pi}})d_\pi \right\|_1 \\
&\leq \gamma \sum_{t=0}^\infty \gamma^t \left\| P_{\tilde{\pi}}^t \right\|_1 \left\| (P_\pi - P_{\tilde{\pi}})d_\pi \right\|_1.
\end{aligned}$$

Since the transition matrix, $P_{\tilde{\pi}}$ is a left stochastic matrix (Asmussen, 2003),

$$\|d_\pi - d_{\tilde{\pi}}\|_1 \leq \gamma \sum_{t=0}^{\infty} \gamma^t \|(P_\pi - P_{\tilde{\pi}})d_\pi\|_1$$

$$= \gamma(1-\gamma)^{-1} \|(P_\pi - P_{\tilde{\pi}})d_\pi\|_1 ,$$

we have that

$$\|(P_\pi - P_{\tilde{\pi}})d_\pi\|_1 = \sum_s \left| \sum_{s',a} \left( P(s'|s,a) \left( \pi(a|s) - \tilde{\pi}(a|s) \right) \right) d_\pi(s) \right|$$

$$\leq \sum_{s,s'} \left| \sum_a \left( P(s'|s,a) \left( \pi(a|s) - \tilde{\pi}(a|s) \right) \right) \right| d_\pi(s)$$

$$\leq \sum_{s,s',a} P(s'|s,a) \left| \left( \pi(a|s) - \tilde{\pi}(a|s) \right) \right| d_\pi(s)$$

$$= \sum_{s,a} \left| \left( \pi(a|s) - \tilde{\pi}(a|s) \right) \right| d_\pi(s) \sum_{s'} P(s'|s,a)$$

$$= \mathbb{E}_{s \sim d_\pi} \|\pi(\cdot|s) - \tilde{\pi}(\cdot|s)\|_1$$

$$= 2\mathbb{E}_{s \sim d_\pi} D_{\mathrm{TV}} \left( \pi(\cdot|s), \tilde{\pi}(\cdot|s) \right)$$

$$\leq \mathbb{E}_{s \sim d_\pi} \sqrt{2 D_{\mathrm{KL}} \left( \pi(\cdot|s), \tilde{\pi}(\cdot|s) \right)}$$

$$\leq \sqrt{2 \mathbb{E}_{s \sim d_\pi} D_{\mathrm{KL}} \left( \pi(\cdot|s), \tilde{\pi}(\cdot|s) \right)}.$$

of which the first two inequalities follow from the triangular inequality, the relationship between $D_{\mathrm{TV}}$ and $D_{\mathrm{KL}}$ is deduced by Pinsker's inequality (Csiszar and Körner, 2011), and the last inequality is by Jensen's inequality with concavity.

From the definition of $D(\theta, \tilde{\theta})$ and (2), we could get

$$\delta(\theta, \tilde{\theta}) \leq \|d_\pi - d_{\tilde{\pi}}\|_1 \left\| \Delta_{\theta,\tilde{\theta}} \right\|_\infty$$

$$\leq \frac{\gamma}{1-\gamma} \|(P_\pi - P_{\tilde{\pi}})d_\pi\|_1 \left\| \Delta_{\theta,\tilde{\theta}} \right\|_\infty$$

$$\leq \frac{\gamma}{1-\gamma} \max_s \Delta_{\theta,\tilde{\theta}}(s) \sqrt{2D(\theta, \tilde{\theta})}.$$

The final result follows from squaring both sides

$$\delta(\theta, \tilde{\theta})^2 \leq 2 \left( \frac{\gamma}{1-\gamma} \max_s \Delta_{\theta,\tilde{\theta}}(s) \right)^2 D(\theta, \tilde{\theta})$$

$$\leq 2 \left( \frac{\gamma}{1-\gamma} \zeta_1 \zeta_2 \right)^2 D(\theta, \tilde{\theta})$$

$$= cD(\theta, \tilde{\theta}).$$

$\square$

## C   Additional Empirical Results

For the algorithm performance as summarized in Figure 1, we provide the comparison results for each game in details. We also provide experimental results with 64 actors for interested readers.

Figure 3: Performance comparison of PPO+DA with PPO on 58 Atari games. Each experiment is allowed to run for 2 hours as a limited time.

Figure 4: Performance comparison of PPO+DA with PPO on 58 Atari games, with the number of used actors increased to 64 and running time increased to 4 hours.