[Reviews · NeurIPS 2019]

Reviewer 1



This paper considers model-free discrete-action reinforcement learning, with the agent learning with variants of stochastic Policy Gradient. The paper introduces and discusses the Bregman Divergence, then presents how it can be used to build a policy loss that allows stable and efficient learning. The core idea of the paper, that I found is best shown by Equation 7, is to optimize the policy by simultaneously minimizing the change between pi_t and pi_t+1 and following the policy gradient. The main contribution of the paper is the use of the Bregman Divergence for the "minimizing change between pi_t and pi_t+1" part of the algorithm. The paper is well-written and interesting to read. It nicely follows a single trajectory, that goes from the Bregman Divergence to its use for Policy Gradient, to the derivation of all the gradients and update rules needed to properly implement the algorithm. The paper is quite dense, with the reader required to be fully attentive from start to finish in order not to miss the slightest piece of notation or derivation. But everything is properly described, and it is possible to understand the idea and its implementation. Algorithm 1 provides helpful pseudocode to tie everything together, even though source code would have been highly desirable (from experience, anything related to gradients needs source code, as the tiniest implementation detail can change the exact gradient being computed). My only (relatively minor) concern with the paper is its lack of intuition. In the first paragraph of this review, I mention Equation 7 and how it enlightened the paper for me. However, this intuition is not in the paper, and I don't even know if it is true or if I misunderstood something in the paper. I think that the paper is currently quite intimidating. For instance, in the abstract, the part with "control the degree of off-policiness" could be replaced with something like "ensures small and safe policy updates even with off-policy data". Ideally, mentioning in the abstract that the Bregman Divergence takes into account the difference of state distributions between two policies, and not only the action probabilities, would allow the reader to fully understand the position and impact of the paper from a quick glance.

Reviewer 2



The work is novel and the derivation appears potentially quite interesting and amenable to follow-up investigation. The clarity of presentation could be improved. In the derivation I found it sometimes hard to keep track of what results were novel and which were prior work, and when a prior result is being extended vs. just quoted. In the empirical results the Bregman divergence is argued to improve exploration. The intuition behind this is unclear to me. By encouraging policy updates to not deviate too far from the existing state action density, it appears similar to trust region methods (i.e. improve stability and avoid local optima). Perhaps the authors could discuss how this can also improve exploration (maybe just by avoiding local minima)? The empirical results are only over 2 hours of training time, so it is difficult to compare asymptotic performance with prior work, particular IMPALA. It would be helpful, for at least a few games, to run for longer to understand how this approach performs asymptotically. Additionally, although it is difficult to compare the x-axis (since the ones here are time, while the original PPO paper is frames), many of the PPO baselines seem weaker and less stable than in the original PPO paper (e.g. ice hockey, video pinball).

Reviewer 3



Thank you for the comprehensive author response. I found answers to O1 and Q2 satisfying. I particularly liked the answer to Q2b which I think might be nice to mention in the work when motivating your technique. I also believe that the authors have appreciated the point I was trying to make in Q1. I sincerely hope that code will be released with this paper, facilitating future implementations and algorithm re-use. One thing that’s not mentioned is how this would scale to the continuous control case. I think mentioning this would make your paper more widely read. Additionally upon re-reading section 5.2.2, I think the Schulman citation you want to use in L254 is actually 2017a and not 2015 since it considers and compares multi-step augmented rewards with single-step rewards. The paper could be made slightly more approachable in terms of its framing and derivation and will be increasing my score in the hopes that the authors update their paper to facilitate this. ======= Originality The authors have done a very good job of providing an excellent related work section that places the method amongst both the mirror policy descent literature as well as the entropy regularized literature. It seems that DAPO is a superset of all the methods. My main concerns are listed below: O1: In L189 the authors mention that the multistep return corresponds to the KL augmented return. So then what is the main contribution of this work? If this is stated in L35 then I think it should be pointed out more strongly in the development of the algorithm. Furthermore, I think discussion the relationship of this work with [Equivalence Between Policy Gradients and Soft Q-Learning Schulman et al 2015] or [Bridging the Gap Between Value and Policy Based Reinforcement Learning Nachum et al. 2017] should make this clearer. Quality I highly appreicate the detailed description of the algorithm settings (5.1) and in appendix A. I anticipate that these will be very useful for people replicated this work. Q1: My first issue is with the quality of the questions in the experimental section 5. Though these question seem important and i understand where the authors are coming from, I find that they are not precise enough. For example, starting a question with “What is the effect” for 1,2 is vague. What are we measuring in these cases? My second issue is the reasons in L235 and L243 for why DAPO would to better than PPO and entropy augmentation respectively. I detail these below in Q2: Q2a: L235 states that the method would perform better for games that may have a local minimum + require deeper exploration. I am ok with the exploration claim since it is clear in Fig 1 that in hard exploration games there is an improvement in performance (though it is split 50/50). However, it is not clear which of these games have different local minimum. I encourage the authors to reframe this argument or provide an experiment that specifically tests this. Q2b: L243 states that entropy augmentation alters the learning objective but doesn’t divergence augmentation also modify the learning objective? Clarity C1: I am not sure why the mirror map formulation is necessary? Isn’t it possible to just start from Eqn 7? Minor issues with math notations: C2a: Eqn 1 — Please provide a plain-text description of this statement. C2b: Eqn 2 —I think the discount factor is missing on the RHS. C2c: L156 I think using c for the loss scaling coefficient is confusion since it is already used as the clipping ratio. Other typos: C3a: L200 (extra “that”) C3b: L224 “without otherwise noted” —> “unless otherwise noted” C3c: L256 “We hypothesis” —> “We hypothesize” Figure improvements: C4: it was hard for me to read the x and y labels in Figure 2.

[Author Response · NeurIPS 2019]

**Response to Reviewer 1**    Thank you for your constructive suggestions. We agree that Equation 7 with the divergence of state-action distribution (instead of only action distribution) is the core idea of the paper. The intuition for considering the discrepancy between state distributions, not only in action space, is to take future divergence into account. We agree that the algorithm should be better described as "ensures small and safe policy updates even with off-policy data". The paper will be updated according to your suggestions.

**Response to Reviewer 2**    We will revise the paper accordingly and improve the clarity of presentation.

**Q:** .. the Bregman divergence is argued to improve exploration. The intuition behind this is unclear to me.

**A**: Intuitively, the divergence constraint can control the current policy $\pi_t$ from going too far from previous policy $\pi_{t-1}$ and retain the stochasticity of policy in the early stage. In addition, the divergence on state-action space also considers the discrepancy of policies on future states, thus encourages deeper exploration (line 41). We will investigate and emphasize the intuition of our method in the future version of paper.

**Q:** .. it is difficult to compare asymptotic performance with prior work, particular IMPALA.

**A:** The empirical difference between IMPALA and ours is mostly due to the number of actors used. As our focus is about the "data-scarce" scenarios, where the data generating speed is far more slower than the training speed (line 45), we use only 16 actors (line 432) to generate samples, while IMPALA use 210 (shallow) / 150 (deep) actors for DMLab and PBT for Atari (Sec.5.3.1). In general, higher score can be achieved by using more actors (more computational resource). Results with more actors and longer training time will be provided in the future version of paper.

**Q:** .. PPO baselines seem weaker and less stable than in the original PPO paper (e.g. ice hockey, video pinball).

**A:** The empirical performance may not be directly comparable with that in original PPO paper and IMPALA, due to the number of actors used, hyper-parameters, training infrastructure etc. We provide the empirical performance with 64 actors on IceHockey and VideoPinball below. The effect of number of actors will be discussed in the future version of paper.

**Response to Reviewer 3**    Thanks for the comments and pointing out typos. Source code will be released in the future.

**O1:** Our contribution is KL augmented return for policy optimization, corresponding to KL divergence on state-action space (line 35). The KL augmentation is between $\pi_t$ and $\pi_{t-1}$, which is different from that in [Schulman et al. 2017a] (see line 191) and entropy in [Nachum et al. 2017]. The contribution will be pointed out more clearly and connection with related work will be discussed further in the future version of paper.

**Q1:** The effect of different factors can be measured in various aspects, e.g. by its stability as a proximal method (Sec 5.2.1), exploratory ability as suggested by divergence on future state distribution (Sec 5.2.2), etc. We will revise the questions more precisely and make the statements more clearly in Section 5.

**Q2: a)** The KL augmentation can be seen as a regularizer for proximal method, which can help escape local minimum. **b)** Intuitively, divergence measures the discrepancy from previous policy $\pi_{t-1}$, thus having infinitesimal effect during convergence (as $D(\pi_t||\pi_{t-1}) \to 0$ asymptotically); while entropy can be seen as measuring discrepancy from an uniform policy $\pi_0$ (i.e. $D(\pi_t||\pi_0)$, which may converge to a positive quantity, thus altering the learning objective).

**C1:** The mirror map formulation is related to the work of MPO and MARWIL and provided simply for completeness. We are also happy to remove this and just start from Equation 7.

**C2 & C3:** Thank you for pointing these out. We will update these parts accordingly in the next version of paper.

**C4:** We will elaborate on the meaning of x-y axis in our next version of paper.

[Meta-Review · NeurIPS 2019]

After discussing, the reviewers agree that this paper makes a good contribution to the field. The main concerns are about improving clarity and presenting more intuition for the results, both of which should be done in the revised version. The reviewers would also like to see source code released so that proper comparisons can be done by future researchers. Additionally, if you haven't already I encourage you to take a look at Belousov and Peter (2018) on f-divergence constrained policy improvement, and clarify its relationship to your work.